# Determining crystal structures through crowdsourcing and coursework

Scott Horowitz[1,2,*], Brian Koepnick[3,*], Raoul Martin[1,4,*], Agnes Tymieniecki[1,2], Amanda A. Winburn[5,6], Seth Cooper[7], Jeff Flatten[8], David S. Rogawski[9], Nicole M. Koropatkin[10], Tsinatkeab T. Hailu[1,11], Neha Jain[1], Philipp Koldewey[1,2], Logan S. Ahlstrom[1,2], Matthew R. Chapman[1], Andrew P. Sikkema[12], Meredith A. Skiba[12], Finn P. Maloney[13], Felix R.M. Beinlich[1,14], Foldit Players[†], University of Michigan students[†], Zoran Popović[8], David Baker[3,15,16], Firas Khatib[17] & James C.A. Bardwell[1,2]

We show here that computer game players can build high-quality crystal structures. Introduction of a new feature into the computer game Foldit allows players to build and real-space refine structures into electron density maps. To assess the usefulness of this feature, we held a crystallographic model-building competition between trained crystallographers, undergraduate students, Foldit players and automatic model-building algorithms. After removal of disordered residues, a team of Foldit players achieved the most accurate structure. Analysing the target protein of the competition, YPL067C, uncovered a new family of histidine triad proteins apparently involved in the prevention of amyloid toxicity. From this study, we conclude that crystallographers can utilize crowdsourcing to interpret electron density information and to produce structure solutions of the highest quality.

[1] Department of Molecular, Cellular, and Developmental Biology, University of Michigan, Ann Arbor, Michigan 48109, USA. [2] Howard Hughes Medical Institute, University of Michigan, Ann Arbor, Michigan 48109, USA. [3] Department of Biochemistry, University of Washington, Seattle, Washington 98195, USA. [4] Biophysics Graduate Group, University of California, Berkeley, California 94720, USA. [5] Center for Complex Networks and Systems Research, Department of Informatics, Indiana University, Bloomington, Indiana 47408, USA. [6] Program in Cognitive Science, Indiana University, 1900 E 10th Street, Bloomington, Indiana 47406, USA. [7] Northeastern University, College of Computer and Information Science, Boston, Massachusetts 02115, USA. [8] Department of Computer Science and Engineering, Center for Game Science, University of Washington, Seattle, Washington 98195, USA. [9] Department of Pathology, University of Michigan, Ann Arbor, Michigan 48109, USA. [10] Department of Microbiology and Immunology, University of Michigan, Ann Arbor, Michigan 48109, USA. [11] ProQR Therapeutics NV., 2333 Leiden, The Netherlands. [12] Department of Biological Chemistry and Life Sciences Institute, University of Michigan, Ann Arbor, Michigan 48109, USA. [13] Chemical Biology Doctoral Program, University of Michigan, Ann Arbor, Michigan 48109, USA. [14] Institute of Complex Systems, Cellular Biophysics (ICS-4), Forschungszentrum, D-52428 Jülich, Germany. [15] Institute for Protein Design, University of Washington, Seattle, Washington 98195, USA. [16] Howard Hughes Medical Institute, University of Washington, Seattle, Washington 98195, USA. [17] Department of Computer and Information Science, University of Massachusetts Dartmouth, Dartmouth, Massachusetts 02747, USA. * These authors contributed equally to this work. Correspondence and requests for materials should be addressed to S.H. (email: horowsah@umich.edu) or to F.K. (email: fkhatib@umassd.edu) or to J.C.A.B. (email: jbardwel@umich.edu).
†A list of consortium members appears at the end of the paper.

'Macromolecular refinement against high-resolution data is never finished, only abandoned'[1]. George Sheldrick's statement on the labour-intensive nature of model building and refining crystal structures reflects the difficulty in producing highly accurate models. As a result, ~85% of deposited protein crystal structures contain discernable errors[2]. Unfortunately, as crystal structures are frequently used as the basis of further studies, inaccurate crystal structures can cause significant harm to the scientific process. Continued improvement of crystal structure accuracy therefore remains an important goal within the biology community.

Recently, in a class assignment, we asked 57 undergraduate students to build the structure of a protein, lectin scytovirin[3], using only the model-building program Coot and an electron density map downloaded from the electron density server[4]. Students were not given the amino-acid sequence of the protein, but were provided with the position of the N-terminal amino acid. The students were instructed to build the structure of the protein, residue by residue, into the $2F_o - F_c$ electron density map. Many students expressed appreciation for the puzzle-like quality of the assignment. In addition to learning about protein structure, ~10% of these students improved on the previously published model[4]. One student even generated a structure that ranked in the 100th percentile in both Molprobity clashscore and total score when compared with other structures in its resolution range. These results raised the intriguing possibility that even a relatively small group of amateur model builders could collectively build higher-quality models than a single trained crystallographer. This concept was remarkably reminiscent of ideas recently championed by the online protein-folding computer game Foldit[5].

On the basis of the success of the undergraduate students in improving a published crystal structure, we hypothesized that crystallographic model building through crowdsourcing might result in more accurate crystal structures than those resulting from traditional model-building methods. Thus, we added electron density features to Foldit to allow players to build directly into density. We then administered a competition to determine if crowdsourcing crystal structures could lead to top-notch structural models. The competition showed that Foldit players could build very high-quality crystal structures into electron density maps, opening up a new method for building and refining crystal structures. In addition, solving the structure of the competition's test protein unexpectedly led to the discovery a new family of histidine triad (HIT) proteins potentially involved in preventing amyloid fibre formation.

## Results

**Adding electron density to Foldit**. Foldit is a popular video game that crowdsources protein structure prediction[5], challenging players to discover low-energy protein models by exploring protein conformational space. The newest version of Foldit provides players with electron density maps and the associated protein sequences, and asks players to use experimental data to guide protein folding (Fig. 1; Supplementary Fig. 1). With these new features, players can trim maps around a model and customize features of the electron density map, such as contour level, rendering style and transparency (Supplementary Fig. 1). The standard Foldit score function is supplemented with a fit-to-density term, allowing in-game structure minimization similar to crystallographic real-space refinement[6]. Foldit players are able to view the fit-to-density score for each residue of a model, providing valuable feedback about specific parts of a model that require more attention. As a preliminary test of this feature, we gave an electron density puzzle to the Foldit players that was nearly identical to the lectin scytovirin classroom assignment mentioned above, and found that the Foldit players were also able to improve on the published scytovirin structure (Supplementary Note 1; Supplementary Fig. 2).

**The model-building competition**. We then held a crystallographic model-building competition to compare the effectiveness of different model-building approaches. Five groups of competitors took part in our model-building competition: (1) 469 Foldit players worldwide, (2) two trained crystallographers, (3) 61 undergraduate students in the University of Michigan class MCDB411 (Introduction to Protein Structure and Function) who built the structure as a class assignment, (4) Phenix Autosolve[7,8] and (5) MR-Rosetta[9]. We chose YPL067C, a yeast protein with no significant sequence similarity to any structure in the Protein Data Bank (PDB) (Supplementary Note 2) as the target for our competition. In addition, YPL067C was chosen for its biological interest, as previous studies suggested that YPL067C is involved in preventing amyloid toxicity[10]. Crystals of this protein diffracted to 1.9 Å resolution (Table 1). We asked all human competitors to build the best possible protein structure that they could given the protein sequence, a secondary structure prediction and an experimentally phased, density refined map of YPL067C. The MCDB411 class assignment was conducted similarly to the previous crystallography assignment discussed above, in which undergraduates improved on a published crystal structure[4]. As before, the undergraduates lacked previous model-building experience. In contrast, 54% of the participating Foldit players had attempted to solve early electron density puzzles in Foldit and thus had some experience in Foldit-based model building. Both the trained crystallographers and undergraduates used the model-building and real-space refinement program Coot[11], whereas the Foldit players used the Foldit version released on 14 October 2015. None of the competitors were given any starting points for building, and thus they needed to establish the relationship between the electron density and the protein's sequence.

The various groups used different approaches. The students and trained crystallographers worked independently, generally utilizing large aromatic residues to identify the relationship between sequence and electron density features. The best Foldit solutions, in contrast, came from a group of players working collaboratively, with one player serving as the trailblazer who contributed the majority of the moves towards the creation of the model (Fig. 1), and other players providing detailed structural tweaks and refinements (Supplementary Movie 1). Although the Foldit players also used a large aromatic residue to initially anchor the sequence similar to the Coot users, their building process sampled conformational space very widely (Supplementary Movie 1), unlike the Coot users. The Foldit players used many different types of building tools, including constraint changes (band, freeze and cut actions), automated minimization algorithms (global wiggle, local wiggle and shake actions) and tools to modify secondary structure (rebuild and tweak actions) (Supplementary Fig. 3).

To determine which group produced the best structural models, we used an automated refinement procedure[12] on all the structures, and then compared key crystallographic statistics. These statistics consisted of $R_{free}$, r.m.s.d.'s of bonds and angles, the number and severity of steric clashes (represented by Molprobity clashscore[13]), and Ramachandran outliers. Analysing the $R_{free}$ values, we quickly realized that the Foldit players were at a distinct disadvantage (Supplementary Fig. 4). Whereas the Coot users, Phenix Autosolve and MR-Rosetta were able to exclude regions that had poor quality electron density, the current version of Foldit required that players model the entire

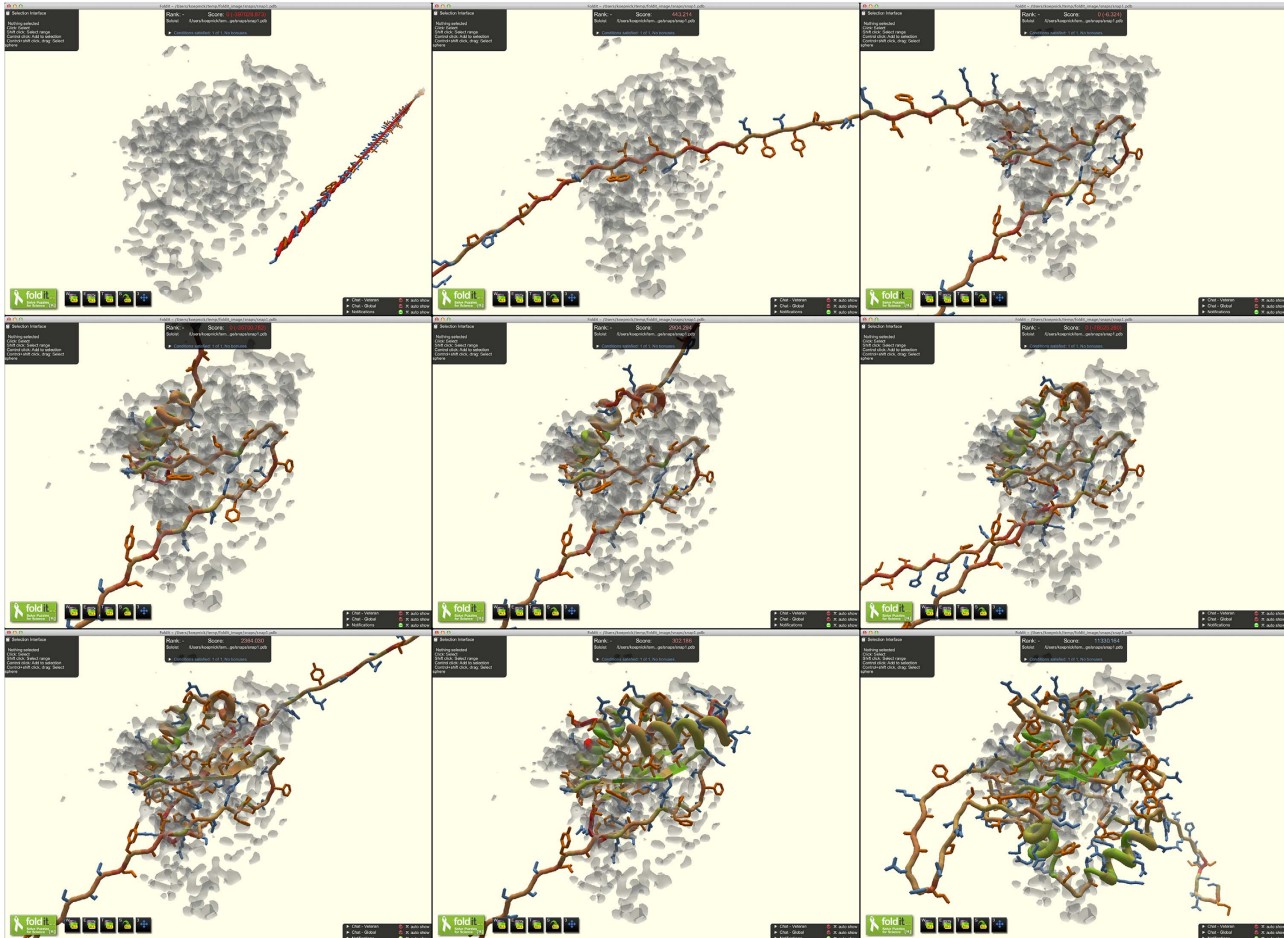

**Figure 1 | Snapshots of Foldit players building YPL067C in the Foldit user interface.** For the complete path of the Foldit model building, see Supplementary Movie 1. The starting point for the puzzle (top left) presented the electron density map and the protein sequence to the player. The players then used Trp108 to help anchor the sequence in electron density (top middle) before beginning to fold secondary structure elements (top right through bottom left). After many rounds of modification in Foldit (bottom middle and Supplementary Movie 1), the players arrived at a high-scoring solution in which the ordered regions of electron density were well fit by YPL067C (bottom right). Disordered regions were later pruned.

sequence without gaps, retaining all disordered residues. Because disordered atoms do not substantially contribute to X-ray diffraction, modelling of disordered residues will disagree with diffraction data and inflate $R_{free}$ values. To correct for this deficiency in Foldit, we pruned the Foldit models afterwards to include only those residues modelled by the top trained crystallographer, so that they contained only the well-ordered regions of the protein. We then re-refined the pruned Foldit structures and compared them with the other models.

The Foldit structures improved considerably after pruning. As a result, the top pruned Foldit structure was the overall highest-quality structure produced in the competition (Fig. 2) as measured by geometry, density fit and steric clashes. In addition to the $R_{free}$ value of the pruned Foldit structure becoming marginally better than that achieved by the trained crystallographers and containing zero Ramachandran outliers, the Foldit structure had the lowest level of steric clashes (Fig. 2). According to Molprobity[13], the top Foldit structure is an exceptional model, ranking in the 100th percentile in both its overall Molprobity score and clashscore of all structures in the PDB of similar resolution (1.95 ± 0.25 Å). The superiority of the top Foldit structure can be attributed to better side-chain conformations than those in the top structure produced by the trained crystallographers (Supplementary Fig. 5). Better Foldit scores were associated with lower $R_{free}$ values (Supplementary Fig. 6),

suggesting that the Foldit model-building strategy and its scoring algorithm could be generalizable as a means of producing high-quality structures.

## Discussion

Here we show that Foldit players can build structural models at least as effectively as trained crystallographers and state-of-the-art automated methods, enabling a novel crowd-powered strategy for solving high-accuracy crystal structures. Combined with the ability to generate molecular replacement solutions in Foldit, and therefore to circumvent the need for experimental phases in some cases[14], it is now potentially possible to obtain complete structure solutions using Foldit given only a native crystallography data set. Citizens hold a tremendous reserve of brainpower that remains largely untapped by the scientific community. The new Foldit electron density feature has revealed that non-expert citizen scientists are capable of using structural data to build first-rate models, and we expect that Foldit will be a powerful tool for crowdsourcing many new high-quality structures.

This surprising win by Foldit players suggests that, in at least some cases, this video game can help produce crystallographic models of higher quality than those from trained crystal-lographers or automated model-building algorithms alone. The difference in accuracy is likely in part the result of different

**Table 1 | Crystallography statistics for HTC1.**

| | SeMet HTC1 (top pruned Foldit) | Native HTC1 |
|---|---|---|
| *Data collection* | | |
| Wavelength (Å) | 0.9876 | 0.97851 |
| Space group | P43212 | P43212 |
| Cell dimensions | | |
| *a, b, c* (Å) | 63.3, 63.3, 117.8 | 62.5,62.5,117.6 |
| *α, β, γ* (°) | 90, 90, 90 | 90, 90, 90 |
| Resolution (Å) | 50–1.95 (1.98–1.95) | 42.81–1.83 (1.89–1.83) |
| $R_{merge}$ (%) | 0.077 (0.940) | 0.074 (0.683) |
| $I/\sigma I$ | 59.5 (1.8) | 12.7 (2.0) |
| Completeness (%) | 99 (89) | 100 (100) |
| Redundancy | 12.4 (7.5) | 7.8 (7.7) |
| Figure of merit | 0.31 | |
| CC1/2 | 0.998 (0.916) | 0.998 (0.873) |
| | | |
| *Refinement* | | |
| Resolution (Å) | 1.95 | 1.83 |
| No. of reflections | 18,107 | 21,161 |
| $R_{work}/R_{free}$ | 0.26/0.28 | 0.20/0.25 |
| No. of non-hydrogen atoms | 1,343 | 1,663 |
| Protein | 1,305 | 1,513 |
| Ligand/ion | 0 | 12 |
| Water | 38 | 138 |
| Average B-factors | 53.8 | 48.8 |
| Protein | 53.8 | 49.0 |
| Ligand/ion | | 51.9 |
| Water | 54.3 | 45.7 |
| R.m.s.d's | | |
| Bond lengths (Å) | 0.008 | 0.009 |
| Bond angles (°) | 1.0 | 0.89 |

SeMet, selenenomethionine.

underlying philosophies behind Coot and Foldit. Whereas Coot primarily uses a real-space refinement system[15] that only respects local geometry, the Rosetta force field used by Foldit is much more extensive, including additional steric, electrostatic and solvation terms, as well as statistical potentials based on observed backbone torsions and side-chain rotamers[6]. That some of the Foldit models were of higher quality than those of trained crystallographers suggests that expert model builders might also benefit from the Foldit force field for real-space refinement. Human intervention either by crystallographers or Foldit players is clearly helpful, as both Phenix Autosolve and MR-Rosetta on their own produced suboptimal structures. The collaborative building process used by the Foldit players could also be a beneficial strategy for professional crystallographers, who could achieve a similar effect by either having multiple laboratory members take turns working on model building and refinement or by submitting their crystal refinement problems to Foldit. Looking forward, we hope that further analysis of electron density puzzle solutions in Foldit can inform continued improvement of automated structure solution algorithms.

Foldit players might also be tasked with improving structures of questionable quality already in the PDB. These Foldit puzzles would benefit the entire community of scientists that depend on accurate structural models. Editing of nearly complete structures could form a base of easier Foldit puzzles for new players, allowing players to practice their model-building skills before moving to more difficult *de novo* model-building puzzles. To further improve the capability for Foldit players to aid crystallographers, ongoing development will make it possible for the players to add or remove residues with insufficient electron

density, and have these changes accurately reflected in the Foldit score. We envision a future in which professional crystallographers frequently tap the collective model-building expertise of Foldit players for help in the model-building, refinement and validation steps of crystallography.

From an educational perspective, the participation of an undergraduate class in this study explored how crystallographic model building can be used not just to teach students the structures and chemistry of proteins in great depth, but in addition, to teach the scientific process. In our previous study, students built into a high-resolution, fully refined map[3]. In the study presented here, students received a lower-resolution, unrefined map and no starting place for building. Dealing with disordered residues, the students were forced to interpret data of varying quality and to decide when the data became too ambiguous to draw firm conclusions. Similar decision-making processes govern the use of scientific data across many disciplines. This assignment thus helped give students a realistic view about the power and limitations of the scientific method. Importantly, the ease with which students and Foldit players were able to interpret and understand density maps suggests that scientists other than crystallographers can very readily interpret electron density maps, which will assist them in designing or analysing experiments based on crystal structures.

YPL067C's structure yielded unexpected insights into its biological function. Despite the lack of sequence homology to any protein in the PDB, a DALI[16] search of YPL067C (Supplementary Table 1) found that it is structurally similar to members of the widely conserved superfamily of histidine triad (HIT) proteins. These proteins contain three histidine residues with an almost identical spatial organization to that of YPL067C, as well as a ß-sheet core nested inside a similar arrangement of loops and helices (Fig. 3). HIT proteins have been shown to be involved in diverse cellular stress responses, such as DNA damage, oxidative stress and induced apoptosis[17,18]. However, the specific *in vivo* activity of HIT proteins remains unclear[17]. Although YPL067C bears some resemblance to known HIT proteins, it is sequentially and structurally distinct (Supplementary Fig. 7). Its most notable distinguishing structural characteristic is an open channel not found in other HIT proteins (Supplementary Note 2). YPL067C's characterization makes it the founding member of a new family we are calling HTC (for histidine triad with channel), with YPL067C being the first member, HTC1. The HTC family contains over 900 members found in a wide variety of eukaryotes and viruses (Supplementary Note 2). As mentioned above, HTC1 null mutants increase the toxicity of amyloid overproduction[10]. We find here that HTC1 is very effective in preventing *in vitro* amyloid formation of three model proteins, $A\beta_{1-40}$, α-synuclein and reduced carboxy-methylated α-lactalbumin (RCMalA; Fig. 4). On the basis of docking simulations, HTC1 may bind to unfolded proteins using its conserved channel (Supplementary Note 2; Supplementary Fig. 8). Our crowdsourcing-enabled discovery of a new family of proteins involved in preventing amyloid formation provides insight into a novel physiological role of the ubiquitous HIT proteins.

## Methods
**Electron density in Foldit.** To facilitate work on electron density data in Foldit, new visualizations and tools, along with a tutorial puzzle to introduce them, were developed and distributed to Foldit players in periodic software updates. Electron density maps in Foldit are displayed as a visual guide in the form of an isosurface. Players have control over parameters of the density isosurface, such as the contour level, surface texture, transparency and colour, and can tag regions of the density with notes. After initial testing, it was clear that density visualization alone was insufficient to improve model building by Foldit players. Players simply ignored the density, finding that their existing, familiar strategies were most competitive on Foldit leaderboards. In response, we adapted the Rosetta fit-to-density score term

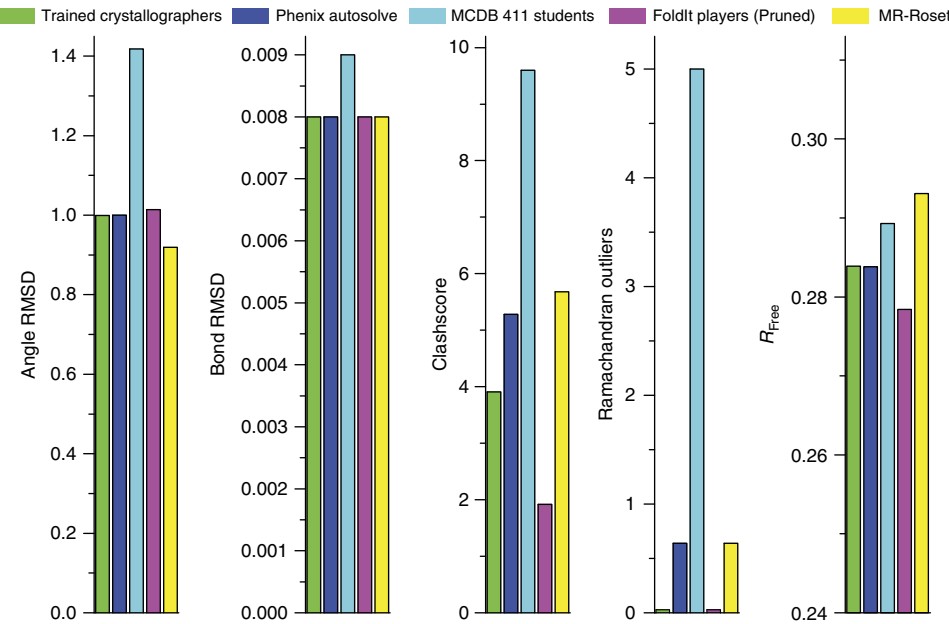

**Figure 2 | Model-building competition results.** Comparison of key statistics of the best model from each group after pruning disordered residues from Foldit structures. In all cases, lower values represent better scores. Comparison before pruning disordered residues is shown in Supplementary Fig. 4.

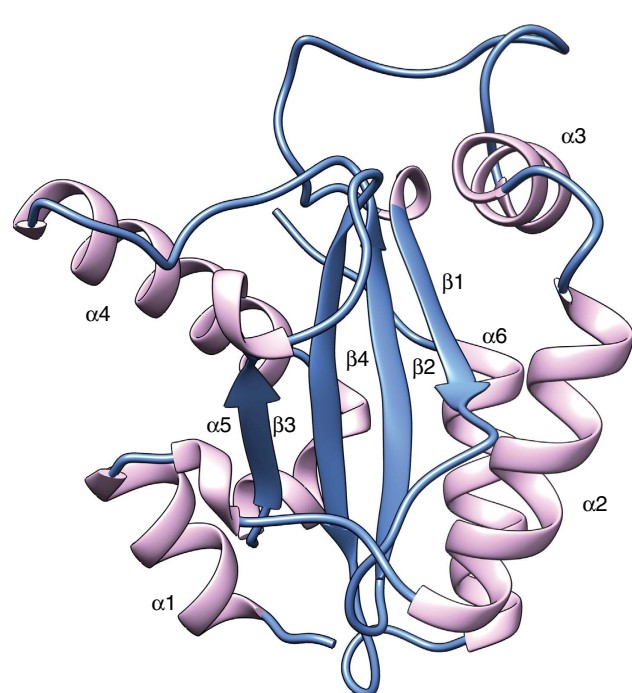

**Figure 3 | Overall structure of HTC1.** Structural alignment with the top DALI search hit is shown in Supplementary Fig. 7.

elec_dens_fast into the Foldit score function[6]. This not only provides competitive incentive to match the density, but also allows Foldit automated tools such as structure minimization to be guided by electron density, similar to crystallographic real-space refinement. Under this configuration, players were able to fit models to several experimental electron density maps with high accuracy. An important feature was added later that allowed players to trim excess density that was distant from the player's model from the visualization. According to Foldit player testimony, this feature has proved invaluable on certain experimental density maps where it allowed a clearer interpretation of relevant density. To protect the integrity of unpublished crystallographic work, electron density data were obfuscated before online distribution to Foldit players.

**The competition.** Phenix Autosolve[7,8], with model-building disabled, was used to create density-modified maps of selenenomethionine (SeMet) YPL067C. To make the map manageable for the Foldit program, the map was masked beyond 5 Å from the initial solution at the start of the competition. This map was given to Foldit players, MCDB411 students and the experienced crystallographers for model building. The individual responsible for model building and refining the initial structure solution of YPL067C, before the contest was initiated, had no contact with any of the competitors.

Sixty-one students in the University of Michigan undergraduate class MCDB411 (Introduction to Protein Structure and Function) were introduced to the assignment through a description of the previous iteration of the assignment in class[4], together with a 1.5 h lecture on X-ray crystallography. Students then had two in-class computer laboratory sessions in which features of Coot were presented. In the first 1.5 h lab session, the students were given basic instructions on opening electron density maps and molecules, changing map levels, scrolling and changing map size, finding secondary structure elements, converting $C_\alpha$ representations to all-atom molecules, placing helices and strands, adding terminal residues, real-space refinement, controlling regularization and refinement, rotating and translating atoms and residues, viewing the skeleton, mutating residues, and changing rotamers. The instructors suggested that changing the weighting of the real-space refinement from the default value of 60–10 and making subsequent changes to this value as needed could help in the building process. In the second 1.5 h lab session, the students were taught how to merge molecules, look for grouped tryptophans, phenylalanines and/or tyrosines as starting places for building, and use validation tools such as density fit analysis, geometry fit analysis and unmodelled blobs. Four instructors were present in the first lab session and three in the second to answer questions on the operation of Coot. Starting from the initial lab session, students were given a total of 1 month to complete the assignment. During this period, the instructor held walk-in help sessions twice a week for 1.5 h each and answered questions on the operation of Coot as well as general model-building questions. Common questions included how to identify density for specific sequences, how to correctly merge molecules and how to approach gaps in electron density. Regarding gaps in density, students were told to model through gaps only if they were confident that the modelling would be correct based on the size of the gap and the number of residues they were modelling in. Students were not told to do in specific cases of building through disordered segments. They were informed that water molecules would not be included in grading. One student asked whether there were external validation tools that could help and was told that the Molprobity server might be useful. Students were allowed to discuss the project and ask each other questions, but were required to do their own model building.

A Foldit puzzle was posted online with the masked electron density map and a model of the target polypeptide in fully extended conformation. Players were challenged to fold the extended polypeptide into the electron density map to achieve a good fit to density. Any advice given to MCDB411 students by the instructors as to how to begin model building was also posted on the Foldit messaging board. After 4 weeks, the puzzle was closed and 900,000-player models were scored and ranked according to the Rosetta energy function. The top scoring

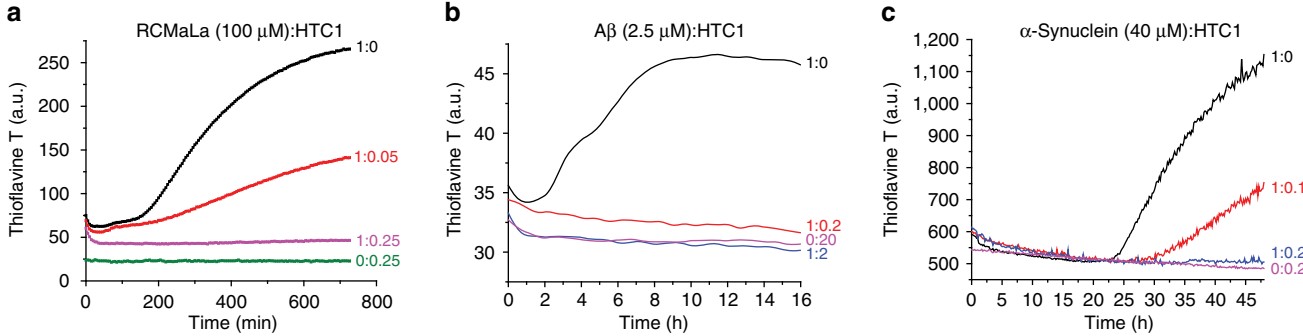

**Figure 4 | HTC1 aggregation inhibition.** HTC1 prevents amyloid formation of RCMLa (**a**) Aβ$_{1–40}$ (**b**) and α-synuclein (**c**), as measured by thioflavin T (ThT) fluorescence at 490 nm.

models were clustered into a set of 1,000 such that no two aligned to $<1.0$ Å $C_\alpha$ r.m.s.d. To this clustered set, we added the 50 best unique models produced by Foldit teams or soloists, as well as any models flagged by Foldit players for special consideration—1,094 Foldit models in total. The puzzle was open to Foldit players for 28 days. Members of the winning team began playing the day after the puzzle opened, and produced the winning structure ∼23 days later—4 days before the puzzle closed.

Two trained crystallographers were given the same number of days for model building as the students and Foldit players. They were given specific instructions not to use tools outside of Coot or Molprobity and not to interact with each other during model building. The trained crystallographers spent ∼8 and 14 h, respectively, working on the puzzle. The trained crystallographers reported using the following approach to the puzzle, which corresponds well with what the instructor observed with many of the undergraduate students. First, they looked for large density blobs that might correspond to large aromatic side chains such as Trp, Tyr or Phe. Working forwards and backwards from the Trp–Phe–Val–Asn sequence proved particularly useful. Modelling in a few of these large residues led to the assignment of density to sequence location. The direction of the polypeptide chain was reversed on a few occasions, but was fixed by looking at the carbonyl density. The Find Secondary Structure tool in Coot was used, especially for regions where the density was poor. Real Space Refine Zone was used with the refinement weight set to 20 or 10, based on the instructor's suggestion for building in an unrefined map. Regions where the density was very poor and decisions had to be made about whether to keep trying to build or not proved to be the hardest part of the task. The trained crystallographers reported that at first they did build in these sections of poor electron density. However, when they realized the extent of the guessing involved, they subsequently removed most of the model in these areas. After modelling in the residues, the trained crystallographers used the validation tools in Coot, including Ramachandran plot, Rotamer analysis and Density Fit analysis, which flagged areas with poor geometry. They also ran the structure through MolProbity, which gave similar results to the Coot validation tools. Finally, the crystallographers fixed problem areas as best as possible with the Coot modelling tools, such as Flip Peptide, Rotamers, Regularize Zone and Real Space Refine Zone. When asked to describe the difficulty level of this assignment, the trained crystallographers rated it as somewhat difficult (on a scale of: very difficult, somewhat difficult, neither easy nor difficult, somewhat easy and very easy).

Phenix Autosolve[7,8] was run with default parameters (using phase_and_build) to produce the Autosolve model. The MR-Rosetta model was obtained by relaxing and rebuilding the Autosolve model in the same electron density map provided to human groups, using Rosetta mr_protocols[9] with nstruct = 10 and selecting the model with the lowest $R_{free}$. ARP/Warp[19] and Phenix Autobuild[20] did not create models of as high quality as Phenix Autosolve or MR-Rosetta, and were thus not analysed in the competition.

After completion of the competition, all structures were automatically refined using Phenix to analyse the results. The refinement strategy included *XYZ* coordinates, temperature factors and updating waters. Notably, the best structures from Foldit, as measured by $R_{free}$, came from the group of highest-scoring Foldit structures according to Foldit score.

**Bioinformatics.** YPL067C sequence conservation was analysed using a four-iteration PSI-BLAST of the UniRef50 database, with an E-value cutoff of 0.005. No sequences in the PDB were found. Sequence conservation was projected onto the structure of YPL067C using the Consurf server[21]. The top DALI[16] match to the crystal structure of YPL067C was to a HIT protein of unknown function from *Clostridium difficile* (PDB: 4EGU), with a Z-score of 4.9. The top 47 hits were all HIT proteins with Z-scores ranging from 4.9 to 4.2 (Z-scores $>2.0$ are considered significant). Secondary structure predictions for the competition were generated using PSIPRED[22].

**Protein expression and purification.** The gene for YPL067C was amplified from yeast strain Y2HGold (Clontech) and cloned into a pET28-sumo plasmid using primer 1 (5′-AAATATGGATCCATGCAACAAGATATCGTCAACGATCAC CAG-3′) and primer 2 (5′-AAATATCTCGAGTCAGGCAAGTGGCTCGAAAC C-3′). pET28-sumo-ypl067C was transformed into *Escherichia coli* BL21(DE3) cells.

Cells were grown at 37 °C overnight in 100 ml Luria Broth (containing 100 µg ml$^{-1}$ kanamycin), and 10 ml was used to inoculate 1 litre Luria Broth (containing 100 µg ml$^{-1}$ kanamycin). At early log phase, the temperature was reduced to 20 °C and 0.1 mM isopropyl β-D-1-thiogalactopyranoside was added to induce expression overnight. Cells were collected by centrifugation and resuspended in 100 ml lysis buffer (40 mM Tris, 10 mM sodium phosphate, 400 mM NaCl, 10% glycerol, 10 mM imidazole, pH 8.0) enriched with 1 mg ml$^{-1}$ DNaseI, 1 mM MgCl$_2$ and two tablets of complete EDTA-free protease inhibitor (Roche). Cells were lysed by two French press cycles at 1,300 p.s.i. and centrifuged at 37,000 *g* for 30 min at 4 °C. The supernatant was run through a Ni-HisTrap 5 ml column (GE Healthcare) pre-equilibrated with lysis buffer at a rate of 1.5 ml min$^{-1}$. Following binding, the column was washed with 60 ml lysis buffer. The protein was eluted with 20 ml lysis buffer enriched with 500 mM imidazole. To cleave the sumo-His × 6 tag, 10 µl ULP1 protease (from stock of 50 mg ml$^{-1}$) was added to the eluted solution. A volume of 10 µl β-mercaptoethanol was added and the solution was dialysed overnight in 40 mM Tris, 10 mM sodium phosphate, 400 mM NaCl, 10% glycerol, pH 8.0. To remove the tag, the solution was run through a Ni-HisTrap 5 ml column (GE Healthcare) pre-equilibrated with dialysis buffer at a rate of 1.5 ml min$^{-1}$, and the flowthrough was saved and diluted in eight volumes of 20 mM Tris, pH 8.0. The protein was then run through a HiTrap Q HP 5 ml column (GE Healthcare), and the flowthrough contained $>95\%$ pure YPL067C as measured by SDS–polyacrylamide gel electrophoresis. Before each experiment, YPL067C was exchanged into appropriate buffer. Expression and purification of SeMet YPL067C was performed with the same protocol except a methionine auxotroph variant of *E. coli* BL21(DE3) and SelenoMethionine Medium Complete (Molecular Dimensions) were used.

α-Synuclein was expressed and purified using the protocol described previously[23] with minor modifications. In brief, 1% of the overnight grown culture was transferred in fresh media and induced with 0.8 mM isopropyl β-D-1-thiogalactopyranoside for 4 h after the optical density of the culture reached 0.6. The induced cells were pelleted at 4,000 r.p.m. and resuspended in 25 ml lysis buffer (10 mM Tris, 1 mM EDTA, pH 8). The lysed cells were then boiled at 95 °C for 15–20 min and centrifuged at 11,000 r.p.m. for 20 min. The supernatant was thoroughly mixed with 10% streptomycin sulfate (136 µl ml$^{-1}$) and glacial acetic acid (228 µl ml$^{-1}$) then centrifuged at 11,000 r.p.m. for 30 min. To the clear supernatant, an equal volume of saturated ammonium sulfate was added, and the solution was incubated at 4 °C for 1 h with intermittent mixing. The precipitated protein was separated by centrifugation at 11,000 r.p.m. for 30 min. The pellet was dissolved in equal volumes of absolute ethanol (chilled) and 100 mM ammonium acetate. Finally, the pellet was washed (twice; optional) with absolute ethanol, dried at room temperature and resuspended in 10 mM Tris, pH 7.4. The protein solution was filtered through a 50 kDa cutoff column (AMICON, Millipore) followed by ion-exchange chromatography (Q-sepharose) against a NaCl gradient. The fractions of pure protein eluted at ∼300 mM NaCl were checked on SDS–polyacrylamide gel electrophoresis and the molecular weight was confirmed by mass spectrometry. The pure fractions were pooled and dialysed overnight against buffer (10 mM Tris and 50 mM NaCl, pH 7.4). The concentration of α-synuclein was determined using $\varepsilon_{280} = 5,600$ M$^{-1}$ cm$^{-1}$. The purified α-synuclein was stored at $-80$ °C at a concentration of ∼100 µM until use.

Aβ$_{1–40}$ peptide was purchased from AlexoTech AB (Umeå, Sweden) and prepared as previously described[24]. Aβ$_{1–40}$ peptide was dissolved in 10 mM NaOH to a peptide concentration of 1 mg ml$^{-1}$ and then sonicated for 1 min in an ice bath before dilution in the assay buffer. The preparations were kept on ice.

α-Lactalbumin (aLA) from bovine milk (cat: L6010) and porcine citrate synthase (cat: C3260-5KU) were purchased from Sigma Inc. RCMaLA was prepared as previously described[25]. aLA (500 μM; freshly prepared in water) was incubated with 1 mM dithiothreitol in 0.5 M Tris and 1 mM EDTA, pH 7.0, for 10 min, then 3 mM iodoacetic acid (out of 1 M stock solution in water) was added and the solution incubated for another 30 min. aLA was then dialysed into 50 mM phosphate buffer, pH 7.0, 100 mM KCl, 10 mM MgCl$_2$.

**Protein crystallization.** Native and SeMet YPL067C crystals were grown at 20 °C by vapour diffusion using both sitting (1 μl drops) and hanging drop methods (2 μl drops). Drops were prepared by mixing a 1:1 solution of YPL067C (25 mg ml$^{-1}$) and reservoir solution (5.6–8.1% glycerol, 1.6–2.1 M ammonium sulfate and 0.1–0.2 M Tris). Crystals were cryoprotected by gradually supplementing the drop with glycerol up to 25% and were flash-frozen in liquid nitrogen.

**X-ray crystallography.** Data were collected at the Life Sciences Collaborative Access Team (LS-CAT) beamlines at the Argonne National Laboratory's Advanced Photon Source at 100 K. The data were integrated and scaled using HKL2000. Phases and initial model building of the SeMet derivative were obtained using Phenix AutoSolve[7,8]. Native YPL067C was solved by molecular replacement with the initial SeMet structure. Iterative refinement and model building were performed using Phenix Refine[12] and Coot[11]. Channel size was analysed using the 3V server[26]. Data collection and modelling statistics are shown in Table 1, and a section of the structure in its 2mFo-DFc map shown in Supplementary Fig. 10.

**Fibrillar aggregation assays.** Fibrillar aggregation was monitored by a thioflavine T (ThT) fluorescence assay. ThT is a benzothiazole dye that exhibits enhanced fluorescence specifically on binding to amyloid fibrils. For RCMaLA aggregation experiments, solutions containing 100 μM RCMaLA, YPL067C in varying concentrations and 20 μM ThT were prepared in 50 mM potassium phosphate buffer, pH 7.0, 100 mM KCl and 10 mM MgCl$_2$ (ref. 25). The ThT fluorescence assays with Aβ$_{1-40}$ peptide were performed with 2.5 μM Aβ$_{1-40}$ peptide, YPL067C in varying concentrations and 20 μM ThT in PBS, pH 7.4, 1% dimethylsulphoxide. The fibrillar aggregation of α-synuclein was tested in a solution of 70 μM α-synuclein, YPL067C in desired concentrations and 20 μM ThT in PBS, pH 7.4. For α-syuclein assays, four glass beads were added in each well to induce aggregation.

ThT fluorescence assays were performed with a final volume of 100 μl of the prepared solution in black 96-microwell plates (costar, UV Plate, 96 well) that were sealed to prevent evaporation. ThT fluorescence was measured in a Synergy HT Multi-Mode Microplate Reader (Biotek) at 37 °C, with constant medium shaking. Excitation and emission wavelengths were 440 and 490 nm, respectively. All samples were assayed in triplicate and the assay was repeated twice. Incubation of YPL067C with ThT alone produced no fluorescence increase.

**Docking of α-synuclein and HTC1.** HTC1 was docked against a 200-member NMR ensemble of α-synuclein[27] using ZDOCK 3.0.2 (ref. 28). The top five scoring poses of HTC1 bound to each member of the ensemble were used to generate a contact frequency map of the HTC1:α-synuclein interaction. To determine the contact map, an interaction was assigned to a given residue pair if their Cα–Cα distance was less than or equal to $\lambda \cdot r_{ij}^{min}$, where $\lambda = 1.2$ and $r_{ij}^{min}$ are taken from the mean Cα–Cα distance for residue pairs that form intermolecular contacts in the PDB[29]. For each intermolecular residue pair, we reported the contact probability averaged over the extracted binding poses. To project the contact maps onto the structures of α-synuclein and HTC1 on the same scale, the contact frequency for each residue pair was averaged over all residues.

**Analytical ultracentrifugation.** Sedimentation velocity experiments of HTC1 (Supplementary Fig. 9) were performed using a Beckman ProteomeLab XL-I analytical ultracentrifuge (Beckman Coulter). YPL067C was first dialysed against 20 mM HEPES, pH 7.5, then diluted to a concentration of 20 or 200 μM using the dialysis buffer. Samples were loaded into cells containing standard sector shaped two-channel Epon-centerpieces with 1.2 cm path length (Beckman Coulter) and equilibrated to 22 °C in the centrifuge for at least 1 h before sedimentation. All samples were spun at 48,000 r.p.m. in a Beckman AN-50 Ti rotor (167431.7 g at the centre of the cell), and the sedimentation of the protein was monitored continuously using the interference optics. Data analysis was conducted with SEDFIT (version 14.1)[30], using the continuous c(s) distribution model. The confidence level for the ME (maximum entropy) regularization was set to 0.7. Buffer density and viscosity were calculated using SEDNTERP (http://sednterp.unh.edu/).

**Data availability.** The final model of HTC1 is deposited in the PDB under the code 5KCI. Other models and raw data are available on request.

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

## Acknowledgements

We thank K. Wan for protein purification, Z. Wawrzak for phasing advice, and U. Jakob for reading and editing the manuscript. This work was supported by National Institutes

of Health (NIH) grants GM102829 to J.C.A.B. and GM118651 to M.R.C. D.S.R. was supported by the University of Michigan Chemistry-Biology Interface training program (NIH grant 5T32GM008597) and the University of Michigan Medical Scientist Training Program (NIH grant 5T32GM007863). M.S. was partially supported by a predoctoral fellowship from the Cellular Biotechnology Training Program (T32GM008353). N.M.K. was supported by the University of Michigan Medical School Host Microbiome Initiative. A.S. was supported by the Molecular Biophysics Training Program (GM008270). B.K. acknowledges support by the National Science Foundation Graduate Research Fellowship Program (DGE-1256082). D.B. and J.C.A.B. are Howard Hughes Investigators. This research used resources of the Advanced Photon Source, a US Department of Energy (DOE) Office of Science User Facility operated for the DOE Office of Science by Argonne National Laboratory under contract no. DE-AC02-06CH11357. Use of the LS-CAT Sector 21 was supported by the Michigan Economic Development Corporation and the Michigan Technology Tri-Corridor (grant 085P1000817). The authors would also like to thank all Foldit players and students who participated in the competition, but chose not to be listed as authors.

## Author contributions

The competition was designed by F.K., S.H. and B.K., and was implemented by S.H., B.K., F.K., N.K., D.R., A.S., F.M. and M.S. Foldit design and improvements were carried out by J.F. and S.C. Experiments were designed by J.C.A.B., A.T., T.H., S.H., P.K. and R.M. Experiments were carried out by A.T., P.K., T.H., F.B. and R.M. N.J., and M.R.C. made the experimental reagents. Analysis was performed by B.K., L.A., P.K., F.K., A.A.W., A.T., D.B., S.H. and J.C.A.B. The paper was written by S.H., J.C.A.B., R.M., A.T. and B.K., with assistance from all authors. U.M.S. and F.P. participated in the competition.

The authors would also like to thank all Foldit players and students who participated in the competition, but chose not to be listed as authors.

## Additional information

**Competing financial interests:** The authors declare no competing financial interests.

**How to cite this article**: Horowitz, S. *et al.* Determining crystal structures through crowdsourcing and coursework. *Nat. Commun.* 7:12549 doi: 10.1038/ncomms12549 (2016).

## Foldit players

Ahmet Caglar[18], Alan Coral[18], Alice Elizabeth Jensen[18], Allen Lubow[18], Amanda Boitano[18], Amy Elizabeth Lisle[18], Andrew T. Maxwell[18], Barb Failer[18], Bartosz Kaszubowski[18], Bohdan Hrytsiv[18], Brancaccio Vincenzo[18], Breno Renan de Melo Cruz[18], Brian Joseph McManus[18], Bruno Kestemont[18], Carl Vardeman[18], Casey Comisky[18], Catherine Neilson[18], Catherine R. Landers[18], Christopher Ince[18], Daniel Jon Buske[18], Daniel Totonjian[18], David Marshall Copeland[18], David Murray[18], Dawid Jagieła[18], Dietmar Janz[18], Douglas C. Wheeler[18], Elie Cali[18], Emmanuel Croze[18], Farah Rezae[18], Floyd Orville Martin[18], Gil Beecher[18], Guido Alexander de Jong[18], Guy Ykman[18], Harald Feldmann[18], Hugo Paul Perez Chan[18], Istvan Kovanecz[18], Ivan Vasilchenko[18], James C. Connellan[18], Jami Lynne Borman[18], Jane Norrgard[18], Jebbie Kanfer[18], Jeffrey M. Canfield[18], Jesse David Slone[18], Jimmy Oh[18], Joanne Mitchell[18], John Bishop[18], John Douglas Kroeger[18], Jonas Schinkler[18], Joseph McLaughlin[18], June M. Brownlee[18], Justin Bell[18], Karl Willem Fellbaum[18], Kathleen Harper[18], Kirk J. Abbey[18], Lennart E. Isaksson[18], Linda Wei[18], Lisa N. Cummins[18], Lori Anne Miller[18], Lyn Bain[18], Lynn Carpenter[18], Maarten Desnouck[18], Manasa G. Sharma[18], Marcus Belcastro[18], Martin Szew[18], Matthew Britton[18], Matthias Gaebel[18], Max Power[18], Michael Cassidy[18], Michael Pfützenreuter[18], Michele Minett[18], Michiel Wesselingh[18], Minjune Yi[18], Neil Haydn Tormey Cameron[18], Nicholas I. Bolibruch[18], Noah Benevides[18], Norah Kathleen Kerr[18], Nova Barlow[18], Nykole Krystyne Crevits[18], Paul Dunn[18], Paulo Sergio Silveira Belo Nascimento Roque[18], Peter Riber[18], Petri Pikkanen[18], Raafay Shehzad[18], Randy Viosca[18], Robert James Fraser[18], Robert Leduc[18], Roberto Dominguez Soto[18], Roman Madala[18], Scott Shnider[18], Sharon de Boisblanc[18], Slava Butkovich[18], Spencer Bliven[18], Stephen Hettler[18], Stephen Telehany[18], Steven A. Schwegmann[18], Steven Parkes[18], Susan C. Kleinfelter[18], Sven Michael Holst[18], T.J.A. van der Laan[18], Thomas Bausewein[18], Vera Simon[18], Warwick Pulley[18] & William Hull[18]

[18] Rosetta Commons, University of Washington, Seattle, Washington 98195, USA.

## University of Michigan students

Alexis Lawton[1], Amanda Ruesch[1], Anjali Sundar[1], Anna-Lisa Lawrence[1], Annes Yukyung Kim[1], Antara Afrin[1], Bhargavi Maheshwer[1], Bilal Turfe[1], Christian Huebner[1], Courtney Elizabeth Killeen[1], Dalia Antebi-Lerrman[1], Danny Luan[1], Derek Wolfe[1], Duc Pham[1], Elaina Michewicz[1], Elizabeth Hull[1], Emily Pardington[1],

Galal Osama Galal[1], Grace Chen[1], Grace Sun[1], Halie E. Anderson[1], Jane Chang[1], Jeffrey Thomas Hewlett[1], Jennifer Sterbenz[1], Jiho Lim[1], Joshua Morof[1], Junho Lee[1], Juyoung Samuel Inn[1], Kaitlin Hahm[1], Kaitlin Roth[1], Karun Nair[1], Katherine Markin[1], Katie Schramm[1], Kevin Toni Eid[1], Kristina Gam[1], Lisha Murphy[1], Lucy Yuan[1], Lulia Kana[1], Lynn Daboul[1], Mario Karam Shammas[1], Max Chason[1], Moaz Sinan[1], Nicholas Andrew Tooley[1], Nisha Korakavi[1], Patrick Comer[1], Pragya Magur[1], Quresh Savliwala[1], Reid Michael Davison[1], Roshun Rajiv Sankaran[1], Sam Lewe[1], Saule Tamkus[1], Shirley Chen[1], Sho Harvey[1], Sin Ye Hwang[1], Sohrab Vatsia[1], Stefan Withrow[1], Tahra K. Luther[1], Taylor Manett[1], Thomas James Johnson[1], Timothy Ryan Brash[1], Wyatt Kuhlman[1], Yeonjung Park[1]

